# Solid-State Gas Sensors with Ni-Based Sensing Materials for Highly Selective Detecting NOx

**DOI:** 10.3390/s24227378

**Published:** 2024-11-19

**Authors:** Zhenghu Zhang, Chenghan Yi, Tao Chen, Yangbo Zhao, Yanyu Zhang, Han Jin

**Affiliations:** 1Institute of Micro-Nano Science and Technology & National Key Laboratory of Advanced Micro and Nano Manufacture Technology, School of Electronic Information and Electrical Engineering, Shanghai Jiao Tong University, Shanghai 200240, China; zhenghu_z@sjtu.edu.cn (Z.Z.); yichenghan@sjtu.edu.cn (C.Y.); 2Nanjing Novlead Biotechnology Co., Ltd., Nanjing 211800, China; tao.chen@novlead.com (T.C.); yangbo.zhao@novlead.com (Y.Z.); yuyan.zhang@novlead.com (Y.Z.); 3Medical School, Henan University, Kaifeng 475004, China; 4National Engineering Research Center for Nanotechnology, Shanghai 200241, China; 5Wuzhen Laboratory, Tongxiang 314500, China

**Keywords:** NOx gas sensor, yttria-stabilized zirconia (YSZ), high selectivity, inhaled nitric oxide, Ni-based sensing materials

## Abstract

Precise monitoring of NO_x_ concentrations in nitric oxide delivery systems is crucial to ensure the safety and well-being of patients undergoing inhaled nitric oxide (iNO) therapy for pulmonary arterial hypertension. Currently, NO_x_ sensing in commercialized iNO instruments predominantly relies on chemiluminescence sensors, which not only drives up costs but also limits their portability. Herein, we developed solid-state gas sensors utilizing Ni-based sensing materials for effectively tracking the levels of NO and NO_2_ in the NO delivery system. These sensors comprised of NiO-SE or (NiFe_2_O_4_ + 30 wt.% Fe_2_O_3_)-SE vs. Mn-based RE demonstrated high selectivity toward 100 ppm NO under the interference of 10 ppm NO_2_ or 3 ppm NO_2_ under the interference of 100 ppm NO, respectively. Meanwhile, excellent stability, repeatability, and humidity resistance were also verified for the proposed sensors. Sensing mechanisms were thoroughly investigated through assessments of adsorption capabilities and electrochemical reactivity. It turns out that the superior electrochemical reactivity of NiO toward NO, alongside the NO_2_ favorable adsorption characteristics of (NiFe_2_O_4_ + 30 wt.% Fe_2_O_3_), is the primary reason for the high selectivity to NO_x_. These findings indicate a bright future for the application of these NO_x_ sensors in innovative iNO treatment technologies.

## 1. Introduction

Pulmonary arterial hypertension (PAH) is a complex and fatal emergency vascular disorder that affects more than 100 million people worldwide, and is mainly found in neonates [1,2]. Compared with oral medicine or injection, inhaled nitric oxide (iNO), a pulmonary vasodilator approved by the Food and Drug Administration (FDA), offers several advantages in the PAH treatment such as noninvasive, targeted, fast onset of action, and less severe systemic adverse effects [3,4,5]. This method, depending on the severity of disease, usually involves the continuous administration of NO gas in the range of 100 ppm by nasal plug or mask [6,7]. However, during treatment, the concentration of NO delivered needs to be tightly controlled in the first place; otherwise, it may cause a rebound in pulmonary arterial pressure and endanger the patient’s life [8]. Additionally, a certain amount of low concentration NO_2_ may be generated during the transport of NO, which also should be precisely detected timely because such a toxic gas can directly endanger patients’ health when the concentration is above 3 ppm [9,10].

It is worth noting that commercialized iNO systems generally use chemiluminescence devices to measure NO and NO_2_, which limits portability and leads to very high cost [3,11]. In recent years, with the rapid development of gas sensors, NOx gas sensors based on solid-state YSZ (yttria-stabilized zirconia)-based electrolytes have received extensive attention from researchers due to their high selectivity, desirable sensitivity, and satisfactory stability [12,13,14,15,16,17,18]. Moreover, YSZ-based sensors can be fabricated in planar configuration, greatly simplifying the structure of the sensor and facilitating the development of integration for commercially available products [13,19,20]. However, currently reported mixed-potential NOx gas sensors mainly focus on the detection of NO and NO_2_ in the middle-to-high concentration range of 100–500 ppm in automotive and industrial contexts [21,22,23,24,25], while the iNO treatment instrument normally requires a relatively low NOx detection limit. Furthermore, it is a great challenge to selectively detect 3 ppm NO_2_ with 100 ppm NO interference, tens times the difference in concentration. Herein, we proposed high-performance YSZ-based NOx sensors by screening out optimal sensing materials. The sensing characteristics of the developed sensors were systematically studied, and we compared the sensors’ performance with some typical NOx gas sensors in Table 1. Moreover, the adsorption capabilities and electrochemical reactivity of these sensors were characteristics to clarify the working principle.

## 2. Experimental

### 2.1. Fabrication of Sensors

Figure 1 depicts the schematic components of the planar sensor. Initially, commercial MnO_2_ powder (99% purity, Sigma Aldrich Chemie GmbH, Taufkirchen, Germany) was thoroughly mixed with α-terpineol, and the obtained paste was painted on the YSZ plate (length × width × thickness: 2.75 × 0.5 × 0.1 cm^3^; NIKKATO CORPORATION, Osaka, Japan); after drying at 130 °C for 4 h, the YSZ plate attached with the MnO_2_ layer was calcined at 1400 °C for 2 h in a muffle furnace to form the Mn-based reference electrode (RE). Various commercial metal oxides (Sigma Aldrich Chemie GmbH, Taufkirchen, Germany, with a purity of 99%) were fabricated with a similar procedure and both were calcined at high temperature for 2 h to form the sensing electrodes (SE).

### 2.2. Evaluation of Sensing Characteristics

The fabricated sensor was set in a quartz tube, and alternatively exposed to the base gas (21 vol.% O_2_, N_2_ balance) or the sample gas containing NO, NO_2_ (total flow rate: 100 mL·min^−1^) for 5 min. As the sensors are expected to be applied in iNO treatment devices in which the process of NO generation and transportation is tightly controlled, as a result, there are few interference gases excepted NO_2_ and the concentration of NO_2_ is below 10 ppm in reality. Moreover, the NO at specific concentrations will be directly inhaled by the patients. Taking these factors into account, in such a specific application, the NO tested concentration range was 10–100 ppm in 21 vol% O_2_, N_2_ balance, and the NO_2_ tested concentration range was 1.5–10 ppm in 21 vol% O_2_, N_2_ balance. An adjustable DC power supply (SS-3305D, A-BF, Dongguan, China) controls the Pt heater’s working voltage to provide a high working temperature for the sensors; notably, the temperatures of the experiments mentioned in the following research just represent the temperature of a tiny area (approximately: 0.4 × 0.4 cm^2^) on the sensor’s surface rather than the whole tests chamber. And there is no significant heat spread, so the temperature of transported gas flow is virtually unaffected. The electrochemical signals were recorded by a Data Acquisition/Switch Unit (34970A, Agilent, Nanjing, China) via Pt wire connected, and the potential difference Δ*V* (Δ*V* = *V*_Sample gas_ − *V*_Base gas_) represented the sensor’s response to sample gas. The surface morphology of calcined powders was observed by Scanning Electron Microscope (Gemini 360, Carl Zeiss, Oberkochen, Germany) with an excitation voltage of 15 kV, and the crystalline phase was characterized by X-ray diffraction (D8 DaVinci, Bruker, Berlin, Germany). Finally, microcantilever chips and an integrated Gas Sensing System (LoC-GSS 1000, High-End MEMS Technology, Xiamen, China) were utilized to test the sensing materials’ gas adsorption.

## 3. Results and Discussion

### 3.1. Screening of Nitric Oxide Sensing Materials and Developing a High-Performance NO Sensor

The sensing mechanism of mixed-potential type NO_X_ can be interpreted by mixed potential theory [13], where the electrochemical reactions for NO (NO_2_) and O_2_ occur simultaneously at the SE/YSZ interface as follows.

For NO:
(1)$$2NO+{2O}^{2-}\to 2NO_{2}+4e^{-}\;(\mathrm{Anodic})$$
(2)$$O_{2}+4e^{-}\;\to 2O^{2-}\;(\mathrm{Cathodic})$$

For NO_2_:
(3)$$2O^{2-}\;\to \;O_{2}+4e^{-}\;(\mathrm{Anodic})$$
(4)$$2NO_{2}+4e^{-}\;\to 2NO+2O^{2-}\;(\mathrm{Cathodic})$$

It can be seen that NO and NO_2_ undergo different reactions respectively and the responses to NO and NO_2_ for YSZ-based NO_X_ sensors are opposite to each other. Generally, the response is negative to NO and positive to NO_2_.

Several metal oxides were selected as potential candidates for a NO sensing material, and among the examined SEs, In_2_O_3_−SE not only gave the highest potential value of 100 ppm NO but also a higher potential value of 10 ppm NO_2_, resulting in poor NO selectivity (Figure 2). In contrast, NiO-SE showed both desirable NO sensitivity and selectivity in comparison with other examined materials. Consequently, NiO was predicted to be the optimal material for NO detection due to its higher sensitivity and selectivity.

The SEM image in Figure 3 shows that the NiO powder annealed at 1200 °C was plate-like and relatively compact. According to former research showing that plate−like metal oxides normally exhibit better selectivity to NO [30,31], we anticipated desirable selectivity for NiO toward NO when against NO_2_.

To confirm the practicability of using NiO tracking a high level of NO with high selectivity against 10 ppm NO_2_, sensing characteristics of the YSZ−based gas sensor comprised of NiO-SE vs. Mn−-based RE were studied. Figure 4a depicts the relationship between the sensor’s response and the operating/calcination temperature for the NiO−SE; it can be confirmed that the optimal response signal was observed for the sensor fabricated at 1200 °C. Moreover, it should be noted the operated temperature of YSZ solid electrolyte must be above 300 °C [32]. And it needs to be clarified that when the operating temperature was below 300 °C, the conductivity of YSZ was greatly reduced, which led to poor recovery characteristics and noticeable noise in the sensor’s response curves. Consequently, the optimal operating/calcination temperature was selected at 310 °C and 1200 °C for the following study. As shown in Figure 4b, the response signal of the sensor varied linearly to the logarithm of the NO concentration in the range of 10–100 ppm. According to the repeated response transients of the sensor that were exposed to 100 ppm NO and 10 ppm NO_2_ (Figure 4c), it can be concluded that the sensor gave excellent repeatability in the successive runs and high selectivity to 100 ppm NO (24.7 mV) against 10 ppm NO_2_ (less than 1 mV). In light of the long-term stability, which is a crucial factor in practical applications. A 35-day test to 100 ppm NO was further measured under 310 °C, as shown in Figure 4d, variation in the response value during the tested period was roughly estimated to be less than 3 mV, indicating satisfactory stability of the sensor using NiO-SE and Mn−based RE.

To further justify the NiO-SE sensor’s ability to differentiate NO in a mixture, Figure 5a compared the responses of the sensor to 100 ppm NO interfered with different concentrations of NO_2_. It could be concluded that the response to 100 ppm NO changed little and the NiO-SE sensor showed excellent anti-interference capability. In addition, the sensor may work under relatively high humidity environments in real practical applications, and Figure 5b illustrates the responses and the change in response values of the NiO-SE sensor at 0%, 24%, 67%, and 91% RH separately to 100 ppm NO at 310 °C. The maximum |ΔV| value was 5% under the range of 0–91% RH, proving that the sensor displays good humidity resistance.

### 3.2. Screening of Nitric Dioxide Sensing Materials and Developing a High-Performance NO_2_ Sensor

It is a great challenge to detect low concentration of NO_2_ compared to ten times the concentration of NO, and pure metal oxides as sensing materials are hardly able to realize the research target. Recently, spinel−based oxides have proven to be good candidates for NO_x_ sensing. For instance, NiFe_2_O_4_ was reported to be a potential NO_2_ sensing material but the inadequate detection limit (LOD) down to 5 ppm limits its application in iNO treatment technologies [23,33,34]. In order to improve the sensing behavior of NO_2_, it is believed that adding additives or dopants to NiFe_2_O_4_ would effectively extend the LOD [35,36]. In the meantime, Fe_2_O_3_ was announced that not only exhibited the same trivalent cation as the NiFe_2_O_4_’s AB_2_O_4_ general formula but also demonstrated acceptable capability to detect NO_2_ [37]. Thus, we added Fe_2_O_3_ into NiFe_2_O_4_ in an attempt to enhance the NO_2_ detection capacity.

Figure 6 demonstrated the XRD patterns for NiFe_2_O_4_, Fe_2_O_3_, and (NiFe_2_O_4_ + x wt.% Fe_2_O_3_) composites after calcined at 1200 °C for 2 h. No impurity peaks were observed, confirming the high purity of the products. However, there were no characteristic diffraction peaks of Fe_2_O_3_ for the composite materials; the reason for this could be Fe_2_O_3_’s lower loading content and weak crystallization.

Furthermore, micro-structures of NiFe_2_O_4_, Fe_2_O_3_, and their composite (e.g., NiFe_2_O_4_ + 30 wt.% Fe_2_O_3_) were characterized via scanning electron microscope (SEM) and presented in Figure 7. In comparison with NiFe_2_O_4_ (Figure 7a,b) and Fe_2_O_3_ (Figure 7c,d), a certain amount of grains (with a diameter of around 0.6 μm), which could be assigned to NiFe_2_O_4_, was observed in the NiFe_2_O_4_ + 30 wt.% Fe_2_O_3_ composite. Moreover, it is reasonable to deduce that the formed composite had a more porous and three-dimensional (3D) structure when compared with that of NiFe_2_O_4_. The porous 3D micro-structure would offer more active reaction sites for NO_2_.

Accordingly, sensing behavior to detect 3 ppm NO_2_ and 100 ppm NO for the YSZ−based gas sensor utilizing NiFe_2_O_4_ and its composite-SEs (vs. Mn-based RE) was examined. As shown in Figure 8a, when the amount of Fe_2_O_3_ reaches 30%, the sensor gave superior selectivity and sensitivity to 3 ppm NO_2_ against 100 ppm NO. In other words, the YSZ−based gas sensor using (NiFe_2_O_4_ + 30 wt.% Fe_2_O_3_)-SE is a good candidate for tracking the trace amount of NO_2_. Figure 8b suggests the optimal operating/calcination temperature was 390 °C/1200 °C.

The dependence of the sensing response on the NO_2_ concentration in the range of 1.5–10 ppm was examined and a linear relationship between response value and NO_2_ concentration on a logarithmic scale (Figure 9a). Notably, it can be concluded that after aging for several days (2–3 days), the response signal of the sensor to 100 ppm NO significantly decreased, whereas it maintained a high response value of 3 ppm NO_2_ (Figure 9b). This phenomenon might be caused by the sensing materials’ gas phase catalytic reaction to NO, whereby a certain amount of NO was converted to NO_2_ before arriving at the TPB, and both NO and NO_2_ triggered the electrochemical reaction in reality. Based on the aforementioned Reaction (1)–(4), the measured potential signal was the final response after the response of NO offset the response of NO_2_. However, in early tests, the mixed materials’ properties were not yet stable, and the conversion efficiency of NO was dynamic. After several days of aging, the NO conversion rate stabilized and the sensor’s response to NO was maintained. Figure 9c,d further indicates the sensor’s excellent repeatability and stability as well as ultra-high selectivity to 3 ppm NO_2_.

The selectivity to NO_2_ of (NiFe_2_O_4_ + 30 wt.% Fe_2_O_3_)-SE could be reinforced in Figure 10a. The responses of the sensor only reduced less than 4 mv even when 3 ppm NO_2_ was mixed with 90 ppm NO and it could be concluded that the (NiFe_2_O_4_ + 30 wt.% Fe_2_O_3_)-SE sensor showed excellent selectivity to NO_2_ at 390 °C. The sensing performance of the (NiFe_2_O_4_ + 30 wt.% Fe_2_O_3_)-SE sensor on the variation in the water vapor content (0–91% RH) was also studied, and Figure 10b shows that the sensor’s response values to 3 ppm NO_2_ were hardly affected by the change in humidity, which could be ascribed to the desorption of water at such a high operating temperature.

### 3.3. Study on the Sensing Mechanism

According to the working principle of an electrochemical gas sensor, the target gas initially selectively adsorbs on the surface of the sensing material and then participates in the following electrochemical reaction. In this case, both adsorption capabilities and electrochemical reactivity contribute to the final response selectivity and response value. To understand which step primarily determines the widely concerned selectivity during NO_x_ sensing, we initially used microgravimetric technology [38,39] to compare the NO_x_ adsorption capabilities of each sensing material. In this study, changes in the microcantilever’s vibration frequency directly reflected the difference in the adsorption capabilities of NO and NO_2_.

Figure 11 shows the frequency shift to 100 ppm NO and 10 ppm NO_2_ of NiO, respectively. In repeated tests, the frequency shift to NO was greater than that of NO_2_, implying that NiO preferred to adsorb NO molecules when exposed to the NO_x_ mixture. In a similar manner, the gas adsorption capabilities of NiFe_2_O_4_(+30 wt.% Fe_2_O_3_) were examined and are presented in Figure 12. It is reasonable to conclude that (NiFe_2_O_4_ + 30 wt.% Fe_2_O_3_) exhibits similar adsorption capability to both 3 ppm NO_2_ and 100 ppm NO, since only a minor difference in the frequency shift to NO and NO_2_ was observed.

To further understand the mechanism for the sensors’ different selectivity, polarization curves were measured in the sample gas (3 ppm NO_2_, 10 ppm NO_2_, 10 ppm NO, and 100 ppm NO, in 21% O_2_, with N_2_ balanced) and air, which could reflect the reaction rate for the electrochemical reaction that occurred in TPB (triple-phase boundary). Figure 13a shows that when the bias voltage was 0 mV, the current value difference between various sample gases at different concentrations and air was small, and it can be inferred that NiO-SE had poor electrochemical catalytic activity to NO and NO_2_. On the contrary, an obvious current difference between NO_2_ and NO is given in Figure 13b. Furthermore, a higher current value of NO_2_ suggests that the (NiFe_2_O_4_ + 30 wt.% Fe_2_O_3_)-SE had better electrochemical catalytic activity of NO_2_, compared with NO.

Through a comparative analysis of gas adsorption capabilities and electrochemical reactivity, we can conclude that the response selectivity of the sensor using NiO−SE primarily arises from its strong adsorption affinity for NO (confirmed by Figure 11). Although NiO exhibits similar electrochemical catalytic activity toward both NO and NO_2_ (Figure 13a), it demonstrates a significant difference in the adsorption of these gases. Conversely, for the sensor utilizing (NiFe_2_O_4_ + 30 wt.% Fe_2_O_3_)-SE, the notable selectivity for NO_2_ can be attributed to the high electrochemical reactivity of NiFe_2_O_4_ + 30 wt.% Fe_2_O_3_ with NO_2_, as only a minor difference was observed in its adsorption capabilities (as presented in Figure 12 and Figure 13b).

## 4. Conclusions

To precisely track the level of NOx during iNO treatment, we developed YSZ-based gas sensors comprised of Ni-based SEs and Mn-based RE. These sensors using NiO or (NiFe_2_O_4_ + 30 wt.% Fe_2_O_3_)-SE demonstrated high selectivity, desirable stability, and excellent repeatability to high-level NO or low-level NO_2_. The working principle of these sensors can be summarized that NO preferred to be adsorbed on the surface of NiO, while NiFe_2_O_4_ + 30 wt.% Fe_2_O_3_ composite demonstrated high electrochemical reactivity to NO_2_, resulting in satisfactory sensing characteristics for these sensors in tracking the variation of NO_x_. However, it should be noted that due to the high operational temperature of these sensors, it is necessary to overcome the limitations of high power consumption in practical applications. Additionally, the current response times might not be fast enough for real-time monitoring, and some strategies, such as a porous sensing layer to accelerate the diffusion rates of gases and machine learning for sensors’ response characteristics to recognize the differences in short-term transient response processes, will be further investigated to improve sensors’ efficiency. In summary, these findings mark a bright future for the application of robust and high-performance NO_x_ sensors in innovative iNO treatment technologies.

## Figures and Tables

**Figure 1 sensors-24-07378-f001:**
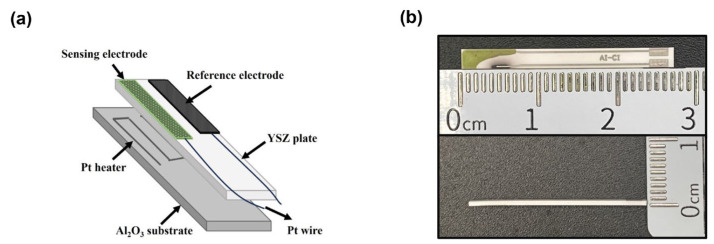
(**a**) Schematic illustration of the planar YSZ-based gas sensor; (**b**) the fabricated miniaturized YSZ-based sensor.

**Figure 2 sensors-24-07378-f002:**
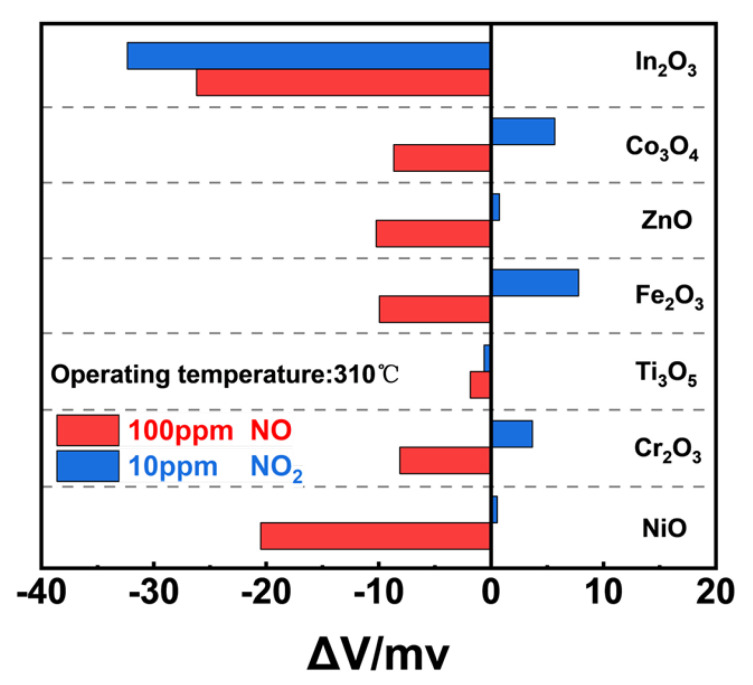
Comparison of sensing responses to 100 ppm NO and 10 ppm NO_2_ in air.

**Figure 3 sensors-24-07378-f003:**
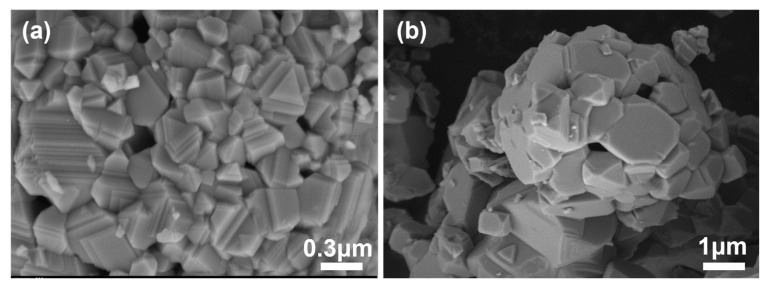
SEM image of NiO powders after calcination with (**a**) high magnification; (**b**) low magnification.

**Figure 4 sensors-24-07378-f004:**
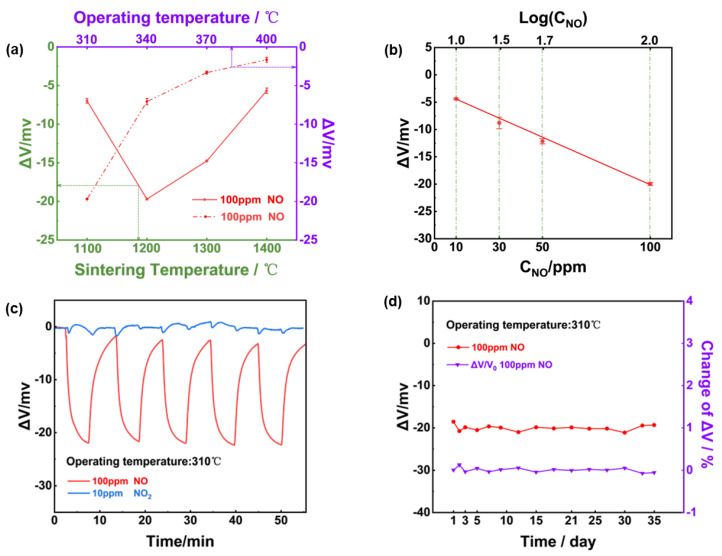
Sensing characteristics of the NiO−SE NO sensor in 21% O_2_ with N_2_ balance: (**a**) variation of the response signal to 100 ppm NO on the operational/calcination temperature; (**b**) dependence of the response signal on the logarithm of the NO concentration in the range of 10−100 ppm at 310 °C; (**c**) repeated response transients to 100 ppm NO and 10 ppm NO_2_ of the sensor at 310 °C; (**d**) long−term stability of the sensor to 100 ppm NO within 35−days.

**Figure 5 sensors-24-07378-f005:**
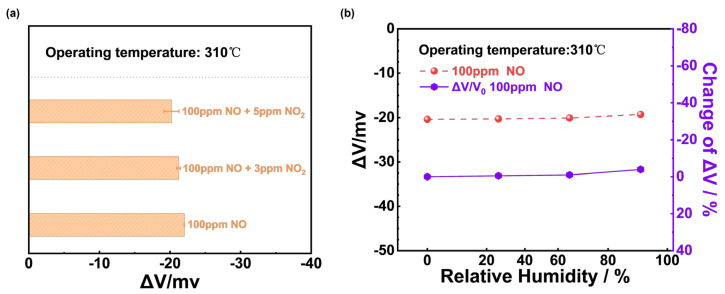
(**a**) Sensing behavior of NiO−SE sensor to 100 ppm NO mixed with different concentrations of NO_2_; (**b**) humidity effect on the response values of NiO−SE sensor under 0−91% RH to 100 ppm NO at 310 °C.

**Figure 6 sensors-24-07378-f006:**
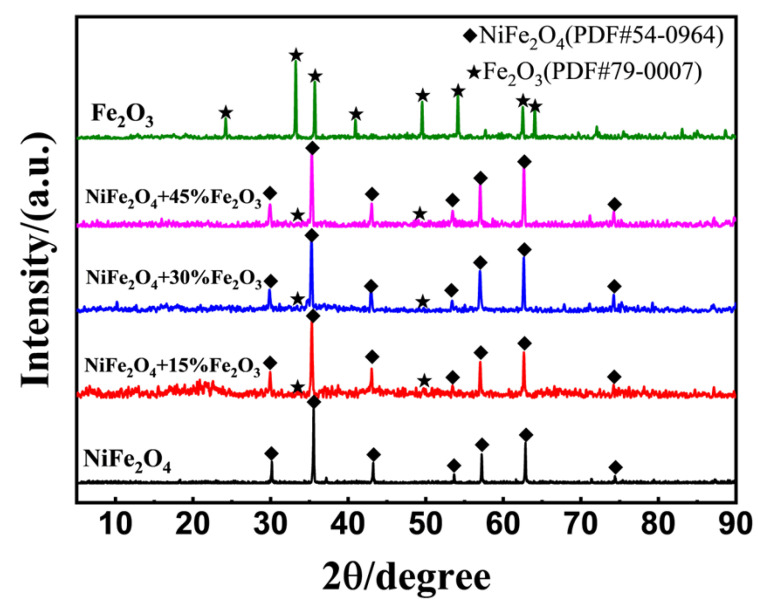
XRD patterns of NiFe_2_O_4_, Fe_2_O_3_, and (NiFe_2_O_4_ + x wt.% Fe_2_O_3_) composites (x = 15%, 30%, 45%), with calcined at 1200 °C for 2 h.

**Figure 7 sensors-24-07378-f007:**
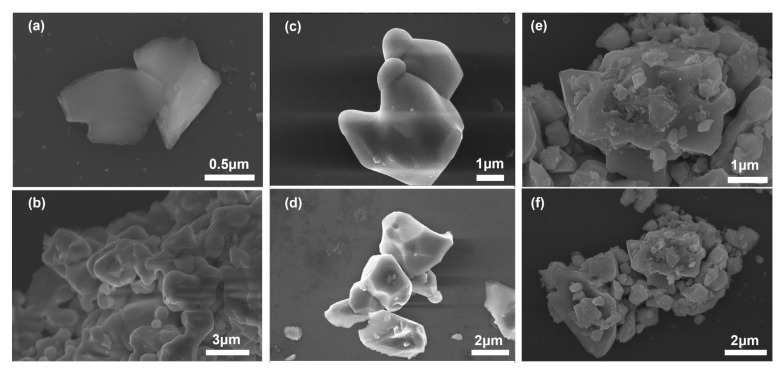
SEM images of (**a**,**b**) NiFe_2_O_4_; (**c**,**d**) Fe_2_O_3_; (**e**,**f**) NiFe_2_O_4_ + 30 wt.% Fe_2_O_3_ composite sensing electrodes prepared at 1200 °C in different scales.

**Figure 8 sensors-24-07378-f008:**
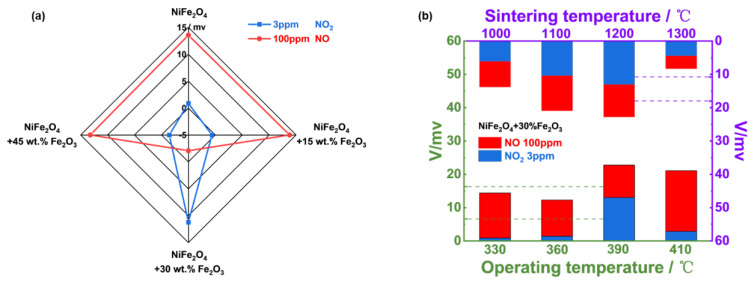
(**a**) Response magnitude to 3 ppm NO_2_ and 100 ppm NO in 21% O_2_ with N_2_ balance respectively for the YSZ-based sensors comprising NiFe_2_O_4_/Fe_2_O_3_ composite SEs and Mn−based RE, operated at 390 °C; (**b**) effect of change in operating/sintering temperature on the sensing characteristic of the sensor using (NiFe_2_O_4_ + 30 wt.% Fe_2_O_3_)-SE in 21% O_2_ with N_2_ balance.

**Figure 9 sensors-24-07378-f009:**
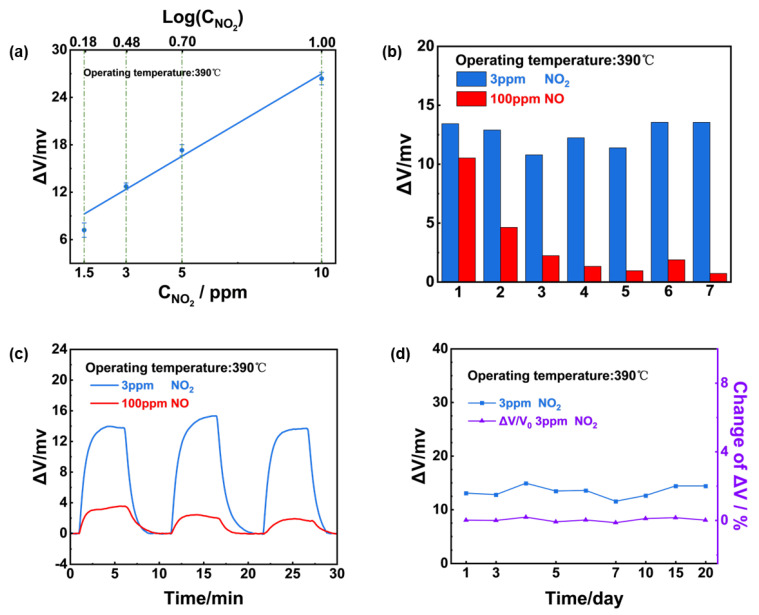
Sensing characteristics of the YSZ−based gas sensor using (NiFe_2_O_4_ + 30 wt.% Fe_2_O_3_)-SE: (**a**) Dependence of the response signal on the logarithm of the NO_2_ concentration in the range of 1.5–10 ppm at 390 °C; (**b**) repeated response to 3 ppm NO_2_ and 100 ppm NO during aging process; (**c**) repeated response transients to 3 ppm NO_2_ and 100 ppm NO; (**d**) long−term stability of the sensor to 3 ppm NO_2_ within 20 day.

**Figure 10 sensors-24-07378-f010:**
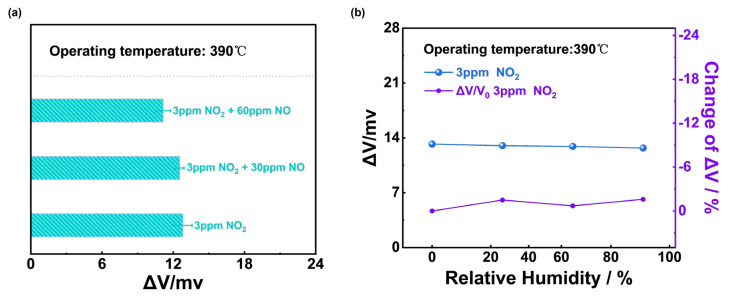
(**a**) Sensing behavior of (NiFe_2_O_4_ + 30 wt.% Fe_2_O_3_)-SE sensor to 3 ppm NO_2_ mixed with different concentrations of NO; (**b**) humidity effect on the response values of (NiFe_2_O_4_ + 30 wt.% Fe_2_O_3_)-SE sensor under 0–91% RH to 3 ppm NO_2_ at 390 °C.

**Figure 11 sensors-24-07378-f011:**
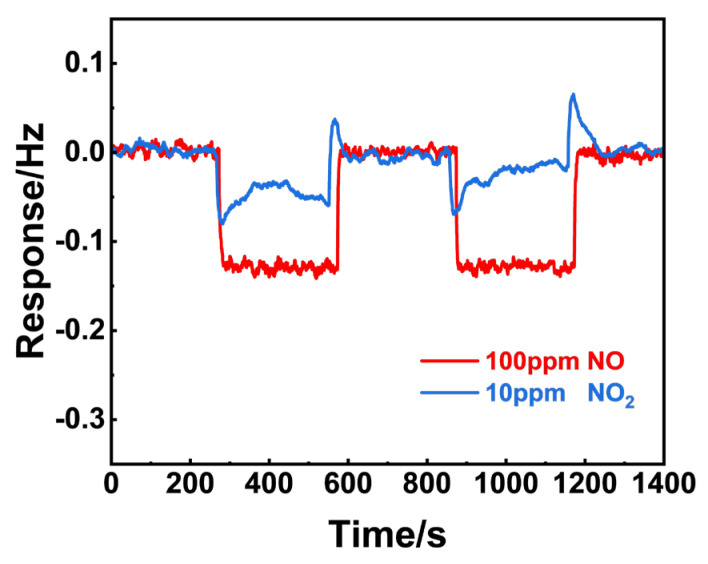
Comparison of the 100 ppm NO and 10 ppm NO_2_ gas adsorption for NiO using the resonant microcantilever.

**Figure 12 sensors-24-07378-f012:**
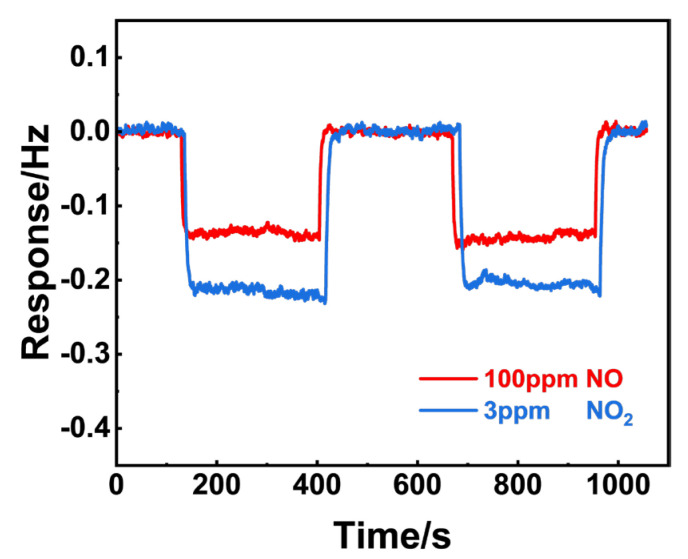
Comparison of the 100 ppm NO and 3 ppm NO_2_ gas adsorption for (NiFe_2_O_4_ + 30 wt.% Fe_2_O_3_) calcined at 1200 °C using the resonant microcantilever.

**Figure 13 sensors-24-07378-f013:**
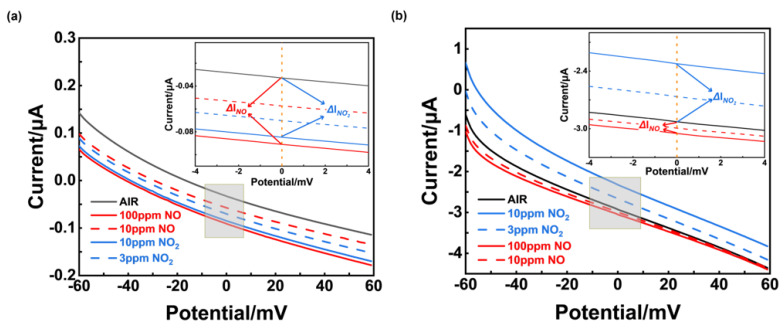
Polarization curves in different concentrations of NO and NO_2_ using different sensors: (**a**) sensor using NiO−SE, operated at 310 °C; (**b**) sensor using (NiFe_2_O_4_ + 30 wt.%Fe_2_O_3_)−SE, operated at 390 °C (ΔI_NO_ and ΔI_NO2_ represent the current value generated from NO or NO_2_, respectively).

**Table 1 sensors-24-07378-t001:** Our work in contrast with the literature.

Sensor Structure SE |YSZ| RE	Temperature (°C)	Detection Range (ppm)	ΔV/Concentration	Ref.
WO_3_ |YSZ| Pt	300	30–500	22 mv/30 ppm NO_2_ (in 5% O_2_)	[22]
NiFe_2_O_4_ |YSZ| Pt	400	100–500	81.3 mv/100 ppm NO_2_ (in 10% O_2_)	[23]
CoTiO_3_ |YSZ| Pt	650	20–200	130 mv/100 ppm NO_2_ (in 5% O_2_)	[24]
La_0_._6_Sr_0_._4_CoO_3_ |YSZ| Pt	400	100–500	125 mv/400 ppm NO_2_ (in 10% O_2_)	[25]
ZnO |YSZ| Pt	700	40–400	−12 mv/100 ppm NO (in 21% O_2_)	[26]
La_2_CuO_4_ |YSZ| Pt	450	50–650	20 mv/650 ppm NO (in 3% O_2_)	[27]
Pt |YSZ pellet| Pt	500	100–800	13 mv/800 ppm NO (in 3% O_2_)	[28]
LCMF |YSZ| Pt	550	——	−62 mv/400 ppm NO (in 1.5% O_2_)	[29]
NiO |YSZ| MnO_2_ NiFe_2_O_4_ + 30%Fe_2_O_3_ |YSZ| MnO_2_	310 390	10–100 1.5–10	−22.7 mv/100 ppm NO 13.8 mv/3 ppm NO_2_ (in 21% O_2_)	this work

## Data Availability

The data presented in this study are available on request from the corresponding author. The data are not publicly available due to the application of patents.

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
