# Peer review of "Solid-State Gas Sensors with Ni-Based Sensing Materials for Highly Selective Detecting NOx"

_sensors, 2024, doi:10.3390/s24227378_

Round 1

Reviewer 1 Report

Comments and Suggestions for Authors

In this study, the authors have fabricated sensors using NiO or NiFe₂O₄ with 30 wt.% Fe₂O₃ as the sensing electrode and a Mn-based reference electrode, demonstrating notable selectivity towards NO and NO₂, respectively. Extensive analyses are presented to establish the sensors' stability, repeatability, and reliability, alongside a mechanistic investigation, which provides an interesting and novel contribution. This work is promising and worthy of publication, but the following issues need to be addressed prior to acceptance:

1.     The abstract states that humidity resistance was verified; however, no specific experiments on humidity tolerance are provided. The reliability of the sensors under varying humidity levels should be experimentally confirmed.

2.     To place the results in context, a performance comparison with other NO sensors reported in the literature is recommended. A comparison table would enhance the clarity of the sensor's advantages and limitations.

3.     Several typographical errors require correction. For example, phrases like “Atomic Force Scanning Electron Microscope” and “frequency shit” need revision.

4.     For NiO sensor, ΔV is observed to be opposite for NO and NO2 (Figure 2). Provide an explanation, as it is important for selectivity.

5.     The plot in Figure 4a suggests that 310 °C may not be the ideal temperature. Additional experimentation at reduced temperatures could provide insights into potentially improved performance at lower temperatures.

6.     Rapid response times are essential to minimize toxic exposure to NOx gases. However, current response times are on the scale of minutes, which is too slow for practical applications. Justify and discuss potential methods for response time improvement.

7.     In Figure 5, peaks specific to Feâ‚‚O₃ are not clearly identified for the composite material. Provide commentary on this observation.

8.     The decrement in NO response over time observed in Figure 8b requires explanation.

9.     In real-world applications, NO and NOâ‚‚ are likely to be present together. The sensor’s ability to differentiate between individual components in a mixture should be discussed and justified. 

Comments on the Quality of English Language

Several typographical errors require correction. 

Reviewer 2 Report

Comments and Suggestions for Authors

This work reports solid-state gas sensors utilizing Ni-based sensing materials to track NO and NO2 levels in the NO delivery system. Although the manuscript reports an interesting approach, several major issues have to be addressed. I will recommend publishing this paper only if all comments are addressed thoroughly.

1) Could you explain why the samples were tested in 21% O2? 

2) The quality of the figures is low. For example, the first image has to be improved or moved to the supporting information. In Figure 11, the inset figures are tiny and hard to read. The font used in the figures should be readable and clear.

3) References should be checked, there are several errors in references. For example, reference 32 is missing.

4) In Figures 9 and 10, since you present a comparison of of the NO and NO2, it's recommended to plot the curves in one graph and not in two separate graphs to make the comparison more reliable and easier to follow up.

5)  The quality of English language and the writing style should be improved. There are several grammatical errors that should be fixed and overall the text should be edited to improve the clarity of the paper.

6) Since the application's aim is to track the level of NOx during iNO treatment, the experiments should be held at a temperature close to the application goals, e.g., 37 C. 

7) In order to claim that the sensor is selective to NO, you should test the sensors with additional gases and compare them to the response to NO to prove that the sensor is selective and responds only to NO. No selectivity tests were performed to prove this claim

Comments on the Quality of English Language

The quality of English and the writing style should be improved. Several grammatical errors should be fixed, and the text has to be edited to improve the paper's clarity.

Round 2

Reviewer 1 Report

Comments and Suggestions for Authors

The authors have adequately addressed all the comments, and the manuscript is now suitable for acceptance in its current form.

Reviewer 2 Report

Comments and Suggestions for Authors

All raised issues have been addressed by the authors